# Phage vB_KlebPS_265 Active Against Resistant/MDR and Hypermucoid K2 Strains of *Klebsiella pneumoniae*

**DOI:** 10.3390/v17010083

**Published:** 2025-01-09

**Authors:** Vyacheslav I. Yakubovskij, Vera V. Morozova, Yuliya N. Kozlova, Artem Yu. Tikunov, Valeria A. Fedorets, Elena V. Zhirakovskaya, Igor V. Babkin, Alevtina V. Bardasheva, Nina V. Tikunova

**Affiliations:** Laboratory of Molecular Microbiology, Institute of Chemical Biology and Fundamental Medicine, Siberian Branch of Russian Academy of Sciences, 630090 Novosibirsk, Russia; morozova@niboch.nsc.ru (V.V.M.); i_babkin@mail.ru (I.V.B.);

**Keywords:** *Klebsiella pneumoniae*, K2-type, molecular serotyping, bacteriophage, phage therapy, genome mosaicism, hypervirulent strain, hypermucoid strain, MDR

## Abstract

*Klebsiella pneumoniae* is an important opportunistic pathogen often resistant to antibiotics. Specific phages can be useful in eliminating infection caused by *K. pneumoniae*. *Klebsiella* phage vB_KlebPS_265 (KlebP_265) and its host strain were isolated from the sputum of a patient with *Klebsiella* infection. KlebP_265 was specific mainly to *K. pneumoniae*-type K2 strains including hypermucoid strains. Most of the hypermucoid KlebP_265-susceptible strains were antibiotic-resistant. This siphophage demonstrated good lytic activity and stability. The KlebP_265 genome was 46,962 bp and contained 88 putative genes; functions were predicted for 37 of them. No genes encoding integrases, toxins, or antibiotic resistance were found in the genome. So, KlebP_265 could potentially be a therapeutic phage. Comparative analysis indicated that KlebP_265 with the most relative *Klebsiella* phage DP01 formed the putative *Dipiunovirus* genus. Genome analysis revealed a large monophyletic group of phages related to KlebP_265 and DP01. This group is divided into two monophyletic clusters of phages forming new putative subfamilies *Skatevirinae* and *Roufvirinae*. Phylogenetic analysis showed extensive gene exchange between phages from the putative subfamilies. Horizontal transfer even involved conservative genes and led to clear genomic mosaicism, indicating multiple recombination events in the ancestral phages during evolution.

## 1. Introduction

The genus Klebsiella belongs to the Enterobacteriaceae family; the members of this genus are encapsulated, immobile, lactose fermenting, facultative anaerobic, and oxidase-negative bacteria that are part of Gram-negative bacilli [1]. This genus is phylogenetically closely related to the genus *Raoultella* and they both form a group called the “Klebsiella clade” [2]. To date, this group includes 22 species (https://lpsn.dsmz.de/genus/klebsiella; https://bacterio.net/genus/raoultella, accessed on 25 September 2024). The type species of the *Klebsiella* genus, *Klebsiella pneumoniae*, is a common opportunistic pathogen included into the ESCAPE group. *K. pneumoniae* can cause a variety of hospital-acquired infections, namely pneumonia, urinary tract infections, and bacteremia in immunocompromised patients or those who have frequent contacts with the healthcare system [3]. The World Health Organization has identified *Klebsiella* strains that have developed resistance to various classes of antibiotics, including multiple drug-resistant (MDR) strains and carbapenem-resistant *Klebsiella* (CRK), as a “critical concern” [4].

*K. pneumoniae* strains are divided into two pathotypes, classical and hypervirulent. Classical strains of *K. pneumoniae* (cKp) are known for the ability to accumulate mutations and acquire genetic determinants leading to the emergence of MDR clones. More than 100 genes of acquired antimicrobial resistance have been identified in *K. pneumoniae*, which encode various products that confer resistance to different classes of antibiotics, including β-lactams, aminoglycosides, quinolones, and polymyxins [5]. Hypervirulent *K. pneumoniae* (hvKp) is more virulent than cKp and can cause severe community-acquired and hospital-acquired infections associated with high pathogenicity and mortality in both immunocompromised and healthy humans [6,7,8,9,10,11]. Initially, hvKp was recognized as a cause of pyogenic liver abscesses in Asia and was identified as an invasive type of *K. pneumoniae* with a tendency to metastatic spread to remote sites, mainly to the eyes, lungs, and central nervous system [6,12]. In addition, severe skin, soft tissue, and bone infections caused by hvKp have been reported [7,13,14,15]. The hvKp infections are diverse and distinct from infections caused by cKp. One of the distinctive features of most hvKp isolates is the hypermucoid phenotype on agar plates. The hypermucoid phenotype is conferred by the *rmpADC* locus and can be detected using genotyping of this *rmp* hypermucoidy locus and the *rmpA2* gene [16]. In addition, the ‘string test’ was developed as a phenotypic test for hypermucoviscosity; when a colony can be stretched by at least five millimeters using an inoculation loop, the isolate is considered hypermucoid and therefore hvKp [17]. Most of the reported hvKp clones belong to serotypes K1 and K2; however, hvKp has also been documented among K5, K16, K54, and K57 serotypes [10]. Most of the previously described hvKp isolates were sensitive to antibiotics. However, hvKp strains are now becoming highly antibiotic-resistant, and thus, the clinical landscape of hvKp is changing drastically [11].

One of the promising approaches to eliminating MDR hvKp infections is the use of phages or phage-derived products. To date, more than 1300 bacteriophages infecting bacteria from the *Klebsiella*/*Raoultella* group are known (https://www.ncbi.nlm.nih.gov/nuccore, accessed on 25 September 2024). The vast majority of the studied phages infect *K. pneumoniae* strains, and only a few phages have been identified whose host strains belong to other *Klebsiella/Raoultella* species. All *Klebsiella* phages are members of the *Caudoviricetes* class, which contains tailed phages with dsDNA genomes. Members of the class can be divided into three morphotypes depending on the characteristics of their tails: myoviruses obtain long tails that are contractile; podoviruses have short noncontractile tails; and siphoviruses have long noncontractile tails. A wide variety of taxonomic groups can be found among *Klebsiella* phages, including 9 families, 14 subfamilies, and 55 genera. *Klebsiella* phages of all three morphotypes of tailed phages were registered, namely siphovirus (e.g., *Casjensviridae*, *Demerecviridae*, *Drexlerviridae*, *Guernseyvirinae*, *Queuovirinae*), myovirus (*Ackermannviridae*, *Peduoviridae*, *Straboviridae*, *Stephanstirmvirinae*, *Vequintavirinae*, *Jedunavirus*), and podovirus morphotype (*Autographiviridae*, *Schitoviridae*) (https://www.ncbi.nlm.nih.gov/taxonomy, accessed on 25 September 2024). However, more than 300 newly isolated *Klebsiella* phages with undetermined systematic position can be found in the NCBI database (https://www.ncbi.nlm.nih.gov/nuccore, accessed on 25 September 2024).

In this study, a novel *Klebsiella* phage vB_KlebPS_265 (hereinafter KlebP_265) was isolated and characterized. Based on its biological properties and the content of its genome, this phage can be suggested as a potentially therapeutic agent. Comparative analysis of the KlebP_265 genome allowed us to propose a new family “*Skateroufviridae*”, which contains two putative subfamilies—“*Skatevirinae*” and “*Roufvirinae*”; the last includes KlebP_265 as a member of a new genus.

## 2. Materials and Methods

### 2.1. Bacterial Strain Identification

*K. pneumoniae* isolate was obtained from a sample of human sputum. Tenfold dilutions of the sample were prepared in sterile phosphate-buffered saline (PBS, pH 7.5), and 100 µL aliquots of cell suspensions were spread on MacConkey agar (Condalab, Madrid, Spain) to differentiate *Enterobacteriaceae* species. Bacterial colonies of interest were transferred to nutrient agar (NA, Microgen, Obolensk, Russia) and spread to obtain individual colonies. One colony was spread on a new NA plate and this procedure was repeated three times to obtain pure culture. Sequencing of a 1300 bp fragment of the 16S rRNA gene was performed to confirm bacterial species. Primers 16s-8-f-B 5′-AGRGTTTGATCCTGGCTCA-3′ and 16s-1350-r-B 5′-GACGGGGGGCGGTGTGTGTGTGTACAAG-3′ were used for PCR and sequencing, as described previously [18]. Sequencing reactions were carried out using BigDye Terminator v.3.1 (Applied Biosystems, Foster City, CA, USA) according to the manufacturer’s instructions. Capillary electrophoresis of the reaction products was performed using an ABI 3500 genetic analyzer (Applied Biosystems, Foster City, CA, USA) and the obtained nucleotide sequences were compared to reference 16S rRNA gene sequences from the NCBI GenBank database (https://www.ncbi.nlm.nih.gov, accessed on 21 September 2023). Bacterial strain was deposited as *K. pneumoniae* CEMTC 5232 (GenBank ID PQ571733) in the Collection of Extremophilic Microorganisms and Type Cultures of the Institute of Chemical Biology and Fundamental Medicine of the Siberian Branch of the Russian Academy of Sciences (CEMTC of the ICBFM SB RAS), Novosibirsk, Russia.

### 2.2. Phage Isolation and Propagation

The phage was isolated from the same clinical sample that was used to isolate its bacterial host. The sputum sample was suspended in 1 mL of sterile PBS, pH 7.5, and sterilized through a 0.22 μm filter (Wuxi Nest Biotechnology, Wuxi, China). Phage plaques were detected by dropping 10 μL aliquots of the filtrate onto a freshly prepared lawn of the *K. pneumoniae* CEMTC 5232 strain in top agar (Becton, Dickinson and Company, Sparks, Difco, Laboratories, Franklin Lakes, NJ, USA). Plates were then incubated overnight at 37 °C. Agar fragments containing phage plaques were cut out from the plates, suspended in sterile PBS, and incubated overnight to extract phage particles. The next day, tenfold dilutions of the phage-containing eluate were applied to a fresh layer of *K. pneumoniae* CEMTC 5232 to obtain single plaques. These plaques were used for subsequent phage isolation, and the cycle of dilution–extraction was repeated three times.

To propagate KlebP_265, *K. pneumoniae* CEMTC 5232 was grown in 50 mL of Lysogenic Broth (LB) (BD Difco, Franklin Lakes, NJ, USA) to an optical density (OD_600_) of 0.3–0.4. Phage particles were then added to exponentially growing *K. pneumoniae* CEMTC 5232 at a multiplicity of infection (MOI) of 0.1, and the infected culture was incubated with shaking at 37 °C until the bacterial lysis occurred. The resulting bacterial lysate was centrifuged at 10,000× *g* for 30 min. Phage particles were purified from the supernatant using polyethylene glycol 6000 (PEG 6000, AppliChem, Darmstadt, Germany) precipitation, as previously described [19]. The phage-containing precipitate was dissolved in 400 μL of STM buffer (0.59 g of NaCl; 7.88 g of Tris-HCl, pH 7.5; and 2.38 g of MgCl_2_ per liter).

### 2.3. Phage Plaques’ and Phage Particles’ Morphology

The morphology of KlebP_265 plaques was determined on a lawn of the host strain *K. pneumoniae* CEMTC 5232. After the plate had been incubated overnight at 37 °C, the plaques were observed. The morphology of phage particles was determined using transmission electron microscopy, with preliminary negative staining, as described previously [20]. The samples were examined using a JEM 1400 transmission electron microscope (JEOL, Tokyo, Japan).

### 2.4. Host Range Analysis of the Phage KlebP_265

The host range was tested against 193 strains of *Klebsiella* spp., which were obtained from the CEMTC, ICBFM SB RAS. The spot test assay was used, as described previously [21]. A bacterial lawn was prepared by mixing 50 μL of an overnight bacterial culture with 4 mL of top agar. Then, 5 μL aliquots of tenfold dilutions of phage preparation were applied to the lawn and allowed to dry. After overnight incubation at 37 °C, the plates were screened for phage plaques. Bacterial strain was considered phage-sensitive if at least a few individual phage plaques were observed on the lawn of the particular strain.

The bacterial susceptibility to KlebP_265 was assessed by the efficacy of plating (EOP) as previously described [22]. The EOP value was calculated as a ratio between the plaque-forming units (PFUs) on the clinical strain tested and the number of PFUs on the isolation/propagation host bacterium.

### 2.5. Molecular Serotyping of K. pneumoniae Strains

K-typing of the strains of *K. pneumoniae* was performed according to the nucleotide sequences of the *wzi* gene, which belongs to the gene cluster encoding synthesis of the polysaccharide capsule. Primers wzi_for 5′-TGCCGCGAGCAGCGCTTTCTATCTTGGTATTCC-3′ and wzi_rev 5′-GAGAGAGCCACTGGTTCCAGAAYTTSACCC-3′ [23] were used for amplification and sequencing of a 580 bp fragment of the *wzi*. The PCR protocol consisted of 30 cycles of amplification, with each cycle including denaturation (94 °C, 30 s), annealing (55 °C, 40 s), and elongation (72 °C, 30 s). Sequencing reactions were carried out using BigDye Terminator v.3.1 (Applied Biosystems, Waltham, MA, USA). The resulting *wzi* sequences were analyzed using K-PAM server (*Klebsiella* species serotype predictor and surface antigen modeler) [24] (www.iith.ac.in/K-PAM, accessed on 20 February 2024).

### 2.6. Hypermucoidity and Antibiotic Resistance of Host Strains of the KlebP_265

Hypermucoidity of *Klebsiella* strains was determined using the string test, as described previously [25]. Bacterial colonies were grown on blood agar base supplemented with 5% sheep blood (OXOID, Basingstoke, Hampshire, UK) at 37 °C overnight. Colonies on the agar media were pulled up using a 10 µL plastic loop, and the viscous string formation was observed. The strain was determined as hypermucous (hm) if a viscous string was >10 mm.

*Klebsiella* strains were tested for sensitivity to antimicrobials using disk diffusion assay (OXOID, Basingstoke, UK) according to the EUCAST recommendations (https://www.eucast.org, accessed on 25 September 2024). The following antibiotics were applied (abbreviation and amount in the disk, µg, are given in parentheses): amoxicillin/clavulanic acid (AMC, 20/10); ampicillin/sulbactam (SAM, 10/10); piperacillin/tazobactam (TZP, 30/10); ticarcillin/clavulanic acid (TIM, 75/10); cefepime (FEP, 30), Cefoxitin (FOX, 30); ceftazidime (CAZ, 10); amikacin (AK, 30); gentamicin (CN, 10); levofloxacin (LEV, 5); ciprofloxacin (CIP, 5); imipenem (IPM, 10); meropenem (MEM, 10); doripenem (DOR, 10); aztreonam (ATM, 30); chloramphenicol (C, 30); trimethoprim/sulfamethoxazole (SXT, 1.25/23.75).

### 2.7. Biological Properties of Phage KlebP_265

The biological properties of the phage KlebP_265 were studied using the *K. pneumoniae* CEMTC 5232 strain as a host. All experiments were performed in triplicate.

Phage adsorption experiments were carried out as described previously [26]. In brief, exponentially growing CEMTC 5232 cells were infected with KlebP_265 at a final concentration of 10^5^ plaque-forming units per milliliter (pfu/mL), and the infected culture was incubated with shaking at 37 °C for 20 min. Aliquots were taken at one-minute intervals to determine the titer of free phages.

To determine the latent period and burst size of the KlebP_265 phage, one-step growth experiments were carried out. An amount of 10 mL of exponentially growing bacteria was centrifuged at 6000× *g* for 5 min, and the bacterial pellet was then resuspended in 500 μL of LB medium (BD Difco, Franklin Lakes, NJ, USA). Phage KlebP_265 was added to the cell suspension at an MOI of 0.001. The culture was incubated for 15 min for phage infection at 37 °C; after that, the cells were pelleted by centrifugation, and the supernatant was removed. The bacterial pellet was suspended in 10 mL of LB. Next, the infected bacterial culture was incubated with shaking at 37 °C for 30 min; aliquots were taken every 2.5 min and used to determine the phage titer.

To construct the multi-step killing curves of bacteria, the experiments were carried out as described previously [27]. Briefly, KlebP_265 phage particles were added to the exponentially growing *K. pneumoniae* CEMTC 5232 at three different MOIs (1, 0.1, and 0.01). The infected cultures were then incubated with shaking at 37 °C for 24 h and aliquots were taken every 30 min to determine the titer of bacteria. The multi-step killing curves were constructed using the data obtained.

To estimate whether the KlebP_265 DNA could integrate into the host genome, the KlebP_265 and CEMTC 5232 cells were co-cultured, resulting in bacteriophage-resistant mutants (BIMs) of the host strain. The experiments were conducted as described previously with some minor modifications [28]. In brief, 10 µL aliquots of the KlebP_265 phage suspension at a titer of at least 10^5^ pfu/mL were dropped on the bacterial lawn of the host CEMTC 5232 in top agar. Then, plates were incubated at 37 °C for 19–20 h. Any colonies of bacteria that grew inside the clear, large plaques were transferred onto NA and tested for their sensitivity to the KlebP_265 phage. To purify BIMs from residual phage particles, they were passaged three times on the plates containing NA and then used to detect KlebP_265 DNA. To confirm the species of the obtained BIMs, 16S rRNA gene sequencing was performed.

Screening for KlebP_265 DNA was carried out using primers KlepP_265_F 5′-CGTCAGTTGTCATAAACCAC-3′ and KlepP_265_R 5′-CAGAATGGAGAGGAAGGATT-3′ (amplify a 586 bp fragment of the ORF26). PCR protocol consisted of 30 cycles of amplification, with each cycle including denaturation (95 °C, 30 s), annealing (54 °C, 30 s), and elongation (72 °C, 1 min). DNA of the KlebP_265 phage and DNA of *K. pneumoniae* CEMTC 5232, respectively, were used as positive and negative controls.

### 2.8. Thermal and pH Stability of KlebP_265

Thermal stability of KlebP_265 was determined as described previously with minor modifications [29]. Phage suspensions in sterile PBS (pH 7.5) were incubated at different temperatures (55 °C, 70 °C, and 90 °C) for 6 h in parallel. Aliquots were taken every hour and the titer of KlebP_265 was determined using the host strain CEMTC 5232. Long-term storage experiments were conducted at different temperatures (−20 °C, 4 °C, 25 °C, 37 °C, and 55 °C) for a period of 135 days in parallel. Aliquots were taken and titers of phage particles were determined at five time points: 24 h, 14 days, 30 days, 45 days, and 135 days. To determine the stability of KlebP_265 at different pH levels, phage suspensions were incubated in sterile TrisHCl buffer (50 mM) at pH values ranging from 2 to 12 for 2 h at room temperature. The residual phage titer was checked using the host strain CEMTC 5232. Thermal and pH stability experiments were performed in triplicate.

### 2.9. Phage DNA Purification and Complete Genome Sequencing

Phage DNA purification was performed as described previously with minor modifications [30]. In brief, phage suspension was mixed with a DNase buffer (Thermo Fisher Scientific, Waltham, MA, USA). DNase I and RNase (both from Thermo Fisher Scientific, Waltham, MA, USA) were then added to the mixture at a final concentration of 5 µg/mL each. The phage-containing solution was incubated at 37 °C for 30 min. Next, EDTA (pH 8.0), SDS, and proteinase K (Thermo Fisher Scientific, Waltham, MA, USA) were added to final concentrations of 20 mM, 0.5%, and 150 µg/mL, respectively. The phage-containing mixture was incubated at 55 °C for 3 h. After that, the KlebP_265 DNA was purified using phenol/chloroform extraction. To precipitate the DNA, 96% ethanol was used, supplemented with 1/30 volume of 3 M sodium acetate (pH 5.1).

A paired-end sequencing library was prepared using NEBNext Ultra II DNA Library Prep Kit for Illumina and Multiplex Oligos for Illumina Dual Index Primers Set 1 (New England Biolabs, Inc., Ipswich, MA, USA) for further sequencing on the MiSeq sequencer with a MiSeq reagent kit v.2 (2 × 250 cycles) (Illumina, San Diego, CA, USA). Trimmomatic v.0.32 [31] (https://github.com/usadellab/Trimmomatic, accessed on 29 September 2024) was used to filter the reads by quality and to remove the adapter sequences. The phage genome de novo assembly was performed using SPAdes v.3.15.4 [32] (http://cab.spbu.ru/software/spades, accessed on 20 December 2023), resulting in one main circular contig with a coverage of 265. The KlebP_265 genome sequence was deposited in GenBank under the accession number PQ480913.

### 2.10. Analysis of Phage Genome

Rapid Automated Annotation System (RAST) v.2.0 [33] (https://rast.nmpdr.org, accessed 14 March 2024) was used to annotate the KlebP_265 genome. Annotation was manually validated using BLASTX search against protein sequences deposited in the NCBI GenBank database (https://ncbi.nlm.nih.gov, accessed on 25 March 2024). In addition, InterProScan and HHpred tools were used to identify protein functions [34,35] (www.ebi.ac.uk/interpro/search, accessed on 27 March 2024; https://toolkit.tuebingen.mpg.de/tools/hhpred, accessed on 28 March 2024). The search for tRNAs was performed using tRNAscan-SE 2.0 software [36] (https://trna.ucsc.edu/tRNAscan-SE, accessed on 27 March 2024). The search for virulence factors and antibiotic resistance genes was carried out using the Virulence Factor (VR) database (http://www.mgc.ac.cn/VFs, accessed on 5 April 2024) and Antibiotic Resistance Gene (AGR) database (https://card.mcmaster.ca/analyze/rgi, accessed on 6 April 2024), respectively. The KlebP_265 genome termini and DNA packaging strategy were determined using the PhageTerm v.1.0.12 tool [37]. SnapGene Viewer software (GSL Biotech; available at https://www.snapgene.com; accessed on 8 April 2024) was used to generate a genomic map. To assess the taxonomy of phage KlebP_265, a comparative proteomic phylogenetic analysis was performed using Viral Proteome Tree Server (ViPTree) [38] (https://www.genome.jp/viptree, accessed on 13 April 2024). Intergenomic similarity (SG) was calculated using Virus Intergenomic Distance Calculator (VIRIDIC) [39] (http://rhea.icbm.uni-oldenburg.de/VIRIDIC, accessed on 13 April 2024).

CoreGenes 5.0 tool [40,41] (https://coregenes.ngrok.io, accessed on 19 April 2024) and Proteinortho6 [42] (https://phage.usegalaxy.eu/?tool_id=toolshed.g2.bx.psu.edu%2Frepos%2Fiuc%2Fproteinortho%2Fproteinortho%2F6.3.1%2Bgalaxy2&version=latest, accessed on 19 April 2024) were used to identify signature genes in the genomes of related phages. Sequences of these signature proteins were then extracted from the NCBI GenBank database (https://blast.ncbi.nlm.nih.gov, accessed on 20 April 2024) and used for further phylogenetic analysis. Protein sequences were aligned and phylogenetic analysis was performed using MEGA X tool [43]. BACPHLIP v. 0.9.3 software [44] was used to predict lifestyle for the phage KlebP_265.

## 3. Results

### 3.1. Phage KlebP_265 Plaques’ and Phage Particles’ Morphology

The KlebP_265 phage produced clear plaques with a diameter of approximately 2 mm, surrounded by a large halo, about 4 mm, on the lawn of the *K. pneumoniae* host strain CEMTC 5232 (Figure 1A). Electron microscopy revealed phage capsids with a diameter of 55.34 ± 1.66 nm, connected to long, flexible tails of 190 ± 3.65 nm in length. The morphology of the phage particles corresponded to that of the siphovirus (Figure 1B).

### 3.2. Biological Properties of KlebP_265

Biological properties of KlebP_265 were investigated using the *K. pneumoniae* strain CEMTC 5232. Phage adsorption experiments revealed that most of the phage particles attach to host cells in 12 min (Figure 2A). The adsorption rate constant was calculated to be 1.8 × 10^−9^ mL/min. Therefore, KlebP_265 can recognize several hundred receptor sites on the cell surface [45]. Phage production was analyzed using a one-step growth assay at an MOI of 0.001. After a latent period of 15 min, phage particles were released from the host cells, and the burst size was determined to be approximately 26 phage particles per cell (Figure 2B).

The lytic activity assay experiments were carried out in three variants with KlebP_265 being added to the cells with an MOI of 0.01, 0.1, and 1 (Figure 2C). Multi-step killing curves of bacteria in the life cycle of the KlebP_265 were calculated, and the obtained data indicated the virulent properties of KlebP_265. It was found that bacterial lysis depends on the dose. An MOI of 1 resulted in the fastest and most significant decrease in the number of viable host cells. In this case, the number of living cells decreased by six orders of magnitude within three hours after infection.

### 3.3. Host Range for the Phage KlebP_265

To determine the host range of the phage KlebP_265, 193 strains of *Klebsiella* spp., including 158 strains of *K. pneumoniae*, were tested. As a result, KlebP_265 was able to infect 25 *Klebsiella* strains (13% of all tested strains), including 1 of the 8 tested strains of *K. aerogenes* and 1 of the 27 tested strains of *K. oxytoca* (Table 1). Notably, all sensitive strains, both *K. pneumoniae* and *K. aerogenes/K. oxytoca*, were isolated from clinical samples collected in several hospitals at different times (Table 1). According to the relative efficiency of plating (EOP), the phage infected *K. pneumoniae* strains more effectively than *K. aerogenes* and *K. oxytoca*. Among the *K. pneumoniae* strains, CEMTC 4128 was the best for phage reproduction, while the lowest level of infectivity was revealed on the lawn of the *K. pneumoniae* CEMTC 2554 strain (Table 1).

It was found that the majority of *K. pneumoniae* strains that were sensitive to the KlebP_265 phage infection belonged to the K2-type (Table 1). Moreover, KlebP_265 was able to infect 60% of the tested K2-type *K. pneumoniae* strains (21 from 35 strains). Two remaining KlebP_265-sensitive *K. pneumoniae* strains, CEMTC 6827 and CEMTC 6848, had the *wzi* allele 173 in their genomes and their K-types could not be clearly determined using only this gene (Table 1). However, KlebP_265 infected only 2 among the 32 tested *K. pneumoniae* strains with this *wzi* allele. Based on the K-PAM database (www.iith.ac.in/K-PAM/, accessed on 20 February 2024), this allele corresponds to three different K-types: KL102, KL149, and KL155 (Table 1), so the K-type of these two strains requires further clarification.

One of the main factors contributing to the pathogenicity of *Klebsiella* is its thick, hypermucous capsule. To identify hypermucoid strains, a string test was conducted using all 193 *Klebsiella* isolates. As a result, 31 isolates were positive for this test, and 12 of them were susceptible to KlebP_265. Eleven of those phage-sensitive hypermucoid strains belonged to the K2-type, and the twelfth was the *K. oxytoca* CEMTC 2335 strain. Notably, it was the only *K. oxytoca* strain that was positive for the string test out of eight studied. It appears that the excessive production of capsule material is one of the factors contributing to the attraction of the KlebP_265 phage (Table 1). The bacterial strains sensitive to KlebP_265 differed in their antibiotic resistance; the host range experiments included *Klebsiella* isolates that were sensitive to all tested antibiotics and those that were resistant to antibiotics from two or more different classes (Table 1 and Appendix A). Despite the fact that the KlebP_265 host strain, *K. pneumoniae* CEMTC 5232, was sensitive to all tested antibiotics and did not possess a hypermucous phenotype, antibiotic-resistant and even MDR strains with increased mucus production were susceptible to KlebP_265.

### 3.4. Phage KlebP_265: Temperature and pH Stability

Experiments on the thermal stability of KlebP_265 revealed that the number of viable phage particles remained constant at 55 °C for 6 h; however, viability decreased rapidly at high temperatures, 70 °C and 90 °C (Figure 3A). Long-term storage experiments indicated that KlebP_265 was stable at various temperatures ranging from 4 °C to 37 °C (Figure 3B). Freezing, however, affected the viability of the phage, causing a decrease in the titer by one order of magnitude after 14 days and by another order of magnitude after 45 days. Storage at 55 °C led to a significant reduction in the titer after one month, with no viable phage particles detected by the end of the 135-day experiment (Figure 3B).

KlebP_265 phage viability was tested at a wide range of pH values, varying from 3 to 10. The phage viability showed no significant difference after 2 h at pH values between 4 and 9. Thus, KlebP_265 is a thermally stable and pH-tolerant phage. The best storage conditions for this phage are at temperatures between 4 °C and 25 °C and at a pH range from 4 to 9.

### 3.5. Annotation and Characterization of the KlebP_265 Genome

The length of the complete genome of KlebP_265 was 46,962 bp with the GC content of 48.07%. PhageTerm analysis indicated no obvious genome termini; therefore, the KlebP_265 phage genome is terminally redundant and completely permuted (Data S1). A total of 88 open reading frames (ORFs) were identified; of these, 34 encoded proteins with predicted functions and 3 corresponded to tRNAs. The remaining 51 ORFs were hypothetical (Figure 4, Appendix A).

Among the ORFs with predicted functions, 13 ORFs encoded DNA metabolism proteins. No DNA- or RNA-polymerase genes were identified in the KlebP_265 genome; however, the genes encoding DNA primase, two DNA helicases, a number of DNA binding proteins, and various endonucleases and exonucleases were determined (Figure 4).

Nineteen ORFs were responsible for virion assembly and DNA packaging (Figure 4). Structural genes encoded a number of capsid and tail proteins. Among them, tail proteins essential for phage attachment and entry into the bacterial cell were found. In particular, ORF26 and ORF28 corresponded to two structural tail proteins with enzymatic activity. The first of them, gp26, is the tail spike protein, which functions as a receptor binding protein. Its endoglycosidase activity is essential for the destruction of the host bacterium capsule during phage infection. The second tail protein, gp28, contains an endopeptidase domain and probably plays an important role in the disruption of the peptidoglycan layer during phage infection. Three other gene products, gp27, gp29, and gp30, were identified as tail tip proteins. Using BLASTP search, gp27 was determined as the TipJ family protein. In addition, HHPred analysis revealed that the N-terminal part of this protein corresponds to the tail tip assembly protein I and its C-terminal part is similar to the tip attachment protein J. Gp29 and gp30 were identified as the Tip L and Tip M proteins. Presumably, these proteins of KlebP_265 form a complex tail tip structure similar to that of the *Escherichia* phage Lambda [46].

Genes encoding tail structures are followed by the gene module associated with capsid morphogenesis. This module contains the genes encoding HK97-gp10-like protein (gp44), major capsid protein (gp56), head decoration/capsid stabilizing protein (gp57), prohead core protein protease (gp58), and some other genes involved in the assembly of viral capsids. In addition, ORF77 and ORF78 are responsible for the head morphogenesis protein and portal protein, respectively. Of the DNA packaging proteins, only the gene of the terminase large subunit, gp88, was identified, but not the terminase small subunit. Probably, it is due to the significant difference in the sequence of the small subunit compared to known sequences. This fact needs further investigation.

Two proteins of the phage lytic cassette have been detected: gp71 and gp72. Gp71 is the Rz-like spanin that is necessary for the destruction of the inner membrane of a bacterial cell, whereas gp72 is a lysozyme (also known as muramidase). No genes that encode integrases, toxins, or antibiotic resistance were found in the genome of KlebP_265. Therefore, this phage meets the criteria for potentially therapeutic phages.

### 3.6. KlebP_265 Lytic Lifestyle Confirmation

To confirm the lifestyle of the phage KlebP_265, the BACPHLIP software was used, which predicted its virulent lifestyle with a probability of 89.86%. In addition, bacteria-insensitive mutants (BIMs) of the host strain were obtained after co-cultivating the host CEMTC 5232 and KlebP_265. PCR for the presence of the ORF27 from the KlebP_265 genome was performed using lysates of these BIMs as a template. No PCR products were detected, indicating that the phage genome was not integrated into the BIM’s genomes. Thus, the *Klebsiella* phage KlebP_265 has a strictly lytic lifestyle.

### 3.7. Comparative Analysis of the Genome and Estimated Taxonomy of the KlebP_265 Phage

KlebP_265 genome was compared with the available phage genomes from the NCBI GenBank database using BLASTN search. It was found that the KlebP_265 genome is most similar at the nucleotide level to the genome of the *Klebsiella* phage DP01 (PP824815) and the nucleotide identity (NI) of these genomes was 0.863. This result was confirmed by the ViPTree tool, which indicated that both genomes formed a common branch on the VipTree dendrogram (Figure 5) and their genomes showed similar gene synteny (Figure 6). In addition, the level of intergenomic similarity (SG) between KlebP_265 and DP01 calculated using VIRIDIC was 87.5% (Figure 7), suggesting that KlebP_265 and DP01 can represent one new genus that can be named *Dipiunovirus* after the DP01 phage. Importantly, both CoreGenes 5.0 and Proteinortho6 revealed 32 core gene products among 34 proteins with predicted functions in these two phages.

Both KlebP_265 and DP01 are members of a monophyletic cluster containing a lot of *Klebsiella* phages with undefined taxonomy [29,47,48,49,50,51,52], the *Aeromonas* phage pIS4-A (NC_042037), which belongs to the *Roufvirus* genus, and the *Vibrio* phage pYD38-A (NC_021534) (Figure 5). To clarify KlebP_265’s taxonomic position, the SG matrix for the KlebP_265 and other phages from this cluster was calculated using VIRIDIC (Figure 7). The VIRIDIC clustering seems to be consistent with the clustering made by ViPTree. In all cases, the SG value varied from 46.1% to 100%, with the exception of the *Salmonella* phage Skate (NC_054639), which was used as an outgroup. Several groups of phages, like KlebP_265 and DP01, showed the SG value > 70% within the groups, which will allow them to be further combined into separate genera (Figure 7). Taking into consideration that the described cluster of phages is monophyletic, contains phages belonging to the approved *Roufvirus* and suggested *Dipiunovirus* genera, and the SG level is more than 46%, this cluster can be proposed as the putative *Roufvirinae* subfamily, denoted after the name of the first approved genus *Roufvirus*. Among the genomes of phages from the putative *Roufvirinae* subfamily, CoreGenes 5.0 and Proteinortho6, respectively, identified 18 and 24 conservative genes encoding proteins with predicted functions, and 6 and 10 conservative genes with hypothetical functions (Table 2). In addition, the analysis of genomes of the putative *Roufvirinae* subfamily revealed that more than half of them encode from one to three tRNA genes. Thus, 12 genomes contained tRNA-Arg-TCT, tRNA-Ile-CAT, or tRNA-Met-CAT, and 5 genomes encoded tRNA-Ser-GCT. The KlebP_265 genome had three tRNA genes, tRNA-Ser-GCT, tRNA-Arg-TCT, and tRNA-Met-CAT.

The neighboring cluster with the proposed *Roufvirinae* subfamily is a monophyletic one containing a number of enterobacterial phages, including *Salmonella*, *Shigella*, and *Escherichia* phages [53,54,55,56,57,58,59,60,61,62,63,64], which are members of several genera (Figure 5). The SG level of the phage genomes, calculated using VIRIDIC, was more than 40% within this cluster (Data S2) and >25% between this cluster and the putative *Roufvirinae* subfamily. Notably, this cluster and the proposed *Roufvirinae* subfamily comprise in turn a monophyletic large group of phages (Figure 5) with similar genome organization (Figure 6). Considering all these facts, it can be supposed this large group of phages represents a new putative family, *Skateroufviridae*, which consists of two putative subfamilies, *Skatevirinae* (named after the most numerous genus) and *Roufvirinae*.

Using CoreGenes 5.0 and Proteinortho6, 13 and 18 genes with predicted functions were found to be conservative between the genomes of phages from the putative *Skateroufviridae* family (Table 2). Importantly, several signature genes, namely major capsid protein, major tail tube protein, and portal protein, as well as TipJ, TipL, and TipM tail tip proteins, which are similar to Lambda phage proteins, were conservative in the phage genomes from both putative subfamilies forming this monophyletic group of phages (Table 2, Figure 5). This indicates their common origin and belonging of phages to the same putative family.

### 3.8. Phylogenetic Analysis of KlebP_265 Proteins

Signature genes are an important characteristic of phages and can be used for taxonomic grouping. Phylogenetic analysis was performed for several signature proteins (both conservative and non-conservative) found in the genomes of phages from the putative *Skateroufviridae* family. The topology of the phylogenetic trees of several core proteins, namely the major tail tube protein gp37 (Figure 8A), prohead core protease gp58 (Appendix A), and InsA C-terminal domain protein gp86 (Appendix A), corresponded to that of the ViPTree (Figure 5); however, dendrograms of other studied signature core proteins (major capsid protein gp56 and portal protein gp78) showed contradictory phylogeny (Appendix A). As for the non-conservative signature proteins, namely terminase, DNA primase, and tail tape measure protein, their dendrograms demonstrated the dissimilar evolutionary history of these proteins (Figure 8B and Appendix A). Notably, the phylogenetic tree of the large subunit terminase, which is one of the most conservative phage proteins and is often used for taxonomy, clearly indicated the horizontal transfer of the terminase genes. The obtained results suggested the mosaicism of the genomes inherent in phages from the putative *Skateroufviridae* family (both *Skateroufvirinae* and *Roufvirinae* subfamilies).

## 4. Discussion

*Klebsiella pneumoniae* is an important opportunistic pathogen often resistant to antibiotics. Specific phages can be useful in eliminating infection caused by *K. pneumoniae.* Recently, a number of *Klebsiella* phages, whose genomes are similar to the investigated KlebP_265, were isolated and characterized [29,47,48,49,50,51,52]. Several of their bacterial hosts have been characterized in terms of K-typing. Thus, phage BUCT541 was active against 1 MDR *K. pneumoniae* K1-ST23 strain [48], phage vB_KpnS_ZX4 reproduced in 1 K1-hv strain of *K. pneumoniae* [52], whereas phage vB_KpnS_SCNJ1-C infected only 1 K54-type strain from 50 tested strains of *K. pneumoniae* [51]. Phage BUCT610 amplified in the *K. pneumoniae* strain K1119f, which belongs to highly virulent ST893 with an undefined capsule type [29]. However, no reported phages from this group were specific to K2 *Klebsiella* strains, which are one of the most common K-types among clinical isolates all over the world [65,66]. So, the investigated phage KlebP_265 is the first phage capable of lysing K2 *Klebsiella* strains; moreover, it has a broad host range, since it was effective against 60% of the tested *K. pneumoniae* K2 strains. Many of the K2 strains acquired MDR phenotype and hypervirulent properties [11]. From the KlebP_265-sensitive K2 strains, several of the hypermucoids were antibiotic-resistant and one was both MDR and hypermucoid. Therefore, this virulent phage could be a promising tool for eliminating *K. pneumonia* infection.

According to previous studies, capsular polysaccharides (CPSs) play an important role in phage adsorption [63,64]. Presumably, K2 capsule polysaccharides interact with the receptor binding proteins (RBPs, tail spikes) of KlebP_265. In particular, the phage is active against hmK2 strains, and this phenotype of *K. pneumoniae* in most cases is associated with hypervirulent characteristics. Moreover, a thick, hypermucous capsule is one of the main factors contributing to the pathogenicity of *Klebsiella*. It was revealed that the KlebP_265 phage preferably infected hmK2 strains of *K. pneumoniae*, as it reproduced in 11 from 12 (92%) tested hmK2 strains. Two mechanisms can be suggested for this phenomenon: the first is that the structure of the hmK2 capsule differs from that of the classical K2-type and the RBPs of KlebP_265 better bind to this structure. The second hypothesis is the probable metabolic difference between cKp and hmKp, which helps the phage to efficiently reproduce in hmKp. Both hypotheses require further investigation.

In addition, KlebP_265 can infect two strains of *K. pneumoniae*, which have 173 allele of the *wzi* gene, 1 *K. aerogenes*, and 1 *K. oxytoca* strain. At the same time, only one gene corresponding to the tail spike protein (gp26) was identified in the phage genome. The tail spike proteins function to bind to receptors on the surface of the microbial cell. Since only one such gene was found in the genome, it is likely that the phage binds to only one type of receptor or structurally similar receptors. Therefore, the capsular polysaccharides of all these strains have structural similarities to the K2-type capsule, which allow the phage tail spikes to attach to the cells of these strains. Note that the phage reproduces in *K. aerogenes* and *K. oxytoca* with a low level of EOP, which indicates its main adaptation to *K. pneumoniae* as a host.

It was found that the phage KlebP_265 is stable in a wide range of temperatures and has good pH tolerance. These properties facilitate the storage of KlebP_265 and make it a promising candidate for use in therapy. It is worth noting that these characteristics are shared by other phages within this group [29,47,48,49,50,51,52].

ViPTree and VIRIDIC analysis based on the complete genomes of KlebP_265 and related phages revealed that KlebP_265 has the closest evolutionary relations with the *K. pneumoniae* phage DP01 and according to the ICTV criteria [67], both phages can be classified as members of a new putative genus *Dipiunovirus*. Moreover, when analyzing related phages using these tools, two putative higher taxonomic groups were identified. The first one unites enterobacterial phages, which belong to eleven genera, namely *Macdonaldcampvirus*, *Saltrevirus*, *Caminolopintovirus*, *Cedarrivervirus*, *Deseoctovirus*, *Swiduovirus*, *Buchananvirus*, *Skatevirus*, *Shuimuvirus*, *Segzyvirus*, and *Akiravirus* (Figure 5). The second cluster contains genomes of a number of *Klebsiella* phages, including KlebP_265 and DP01, *Vibrio* phage pYD38-A (NC_021534), and *Aeromonas* phage pIS4-A (NC_042037). According to comparative genome analysis, both clusters are monophyletic and share intergenomic similarities, as well as gene syntheny within each cluster. So, both clusters can be classified as two separate subfamilies, *Skatevirinae* and *Roufvirinae*, which can be united into one putative new family *Skateroufviridae*. The analysis of signature genes revealed that phages that form the putative *Skatevirinae* and *Roufvirinae* subfamilies have a similar range of core genes, with most of these genes belonging to a structural cluster of tail proteins (Table 2). In addition, thirteen and twenty genes were identified using CoreGene 6.0 and Proteinortho6, respectively, as signature genes for the putative family *Skateroufviridae*.

Importantly, the signature genes encoding the terminase, portal protein, and major capsid protein, which are usually conservative and often used for phage taxonomy [68], were absent among the core genes and phylogenetic analysis demonstrated their involvement in gene exchange between phages from the putative *Skatevirinae* and *Roufvirinae* subfamilies. Multiple horizontal transfers were found for a lot of other genes with predicted functions. This led to extensive genome mosaicism in phages from the suggested *Skateroufviridae* family that is a feature of members from this family. Apparently, mosaicism is described for a number of other phages [69,70,71]. Probably, the observed genomic mosaicism is the result of a number of recombination and rearrangement events that occurred earlier in the genomes of their common ancestors [71,72]. As for members of the putative *Skateviridae* family, this assumption is consistent with the presence of several recombinase and HNH nuclease genes in the KlebP_265 genome and genomes of related phages.

In conclusion, we characterized KlebP_265 as a lytic phage against a spectrum of K2 MDR and hmK2 strains of *K. pneumoniae*. The biological properties and genome characteristics show that the phage KlebP_265 has the potential to effectively eliminate the most common clinical *K. pneumoniae*. According to comparative genome analysis, this phage is a member of a new genus, which in turn is part of a new subfamily and family.

## Figures and Tables

**Figure 1 viruses-17-00083-f001:**
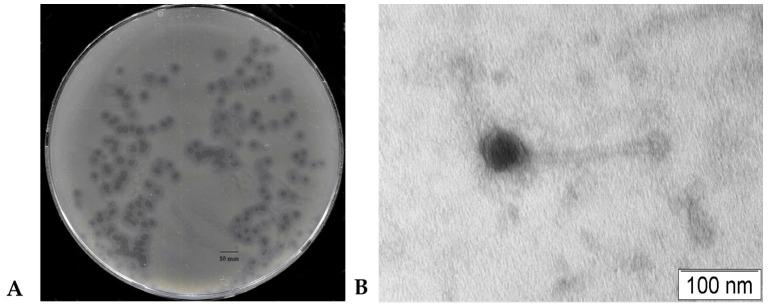
Phage KlebP_265 characteristics. (**A**) Phage plaques’ morphology on the lawn of *K. pneumoniae* host strain CEMTC 5232; (**B**) electron micrograph of the phage KlebP_265.

**Figure 2 viruses-17-00083-f002:**
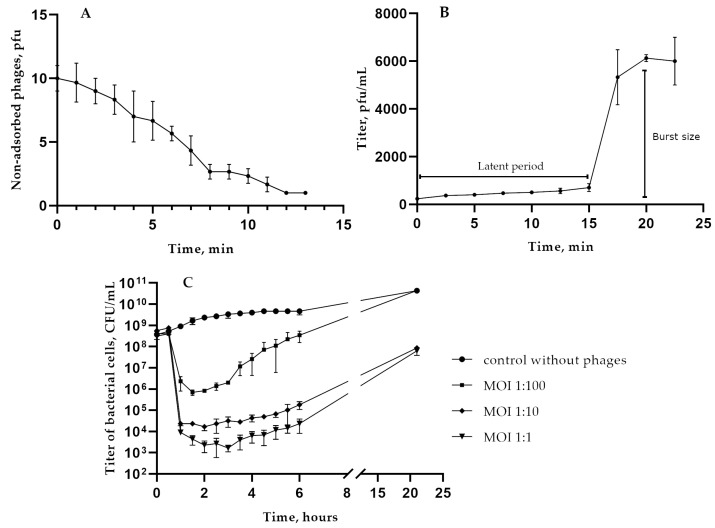
Biological properties of the phage KlebP_265. (**A**) Phage adsorption experiments, (**B**) one-step growth assay, (**C**) multi-step bacterial lytic curves for the host *K. pneumoniae* CEMTC 5232, infected with phages at different MOIs. The bars show standard deviations for each point.

**Figure 3 viruses-17-00083-f003:**
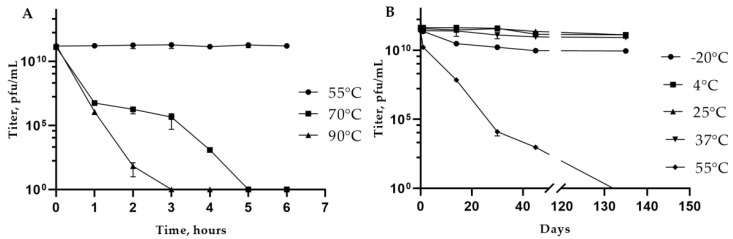
KlebP_265 thermal stability experiments. (**A**) Stability at high temperatures, (**B**) long-term storage of the phage.

**Figure 4 viruses-17-00083-f004:**
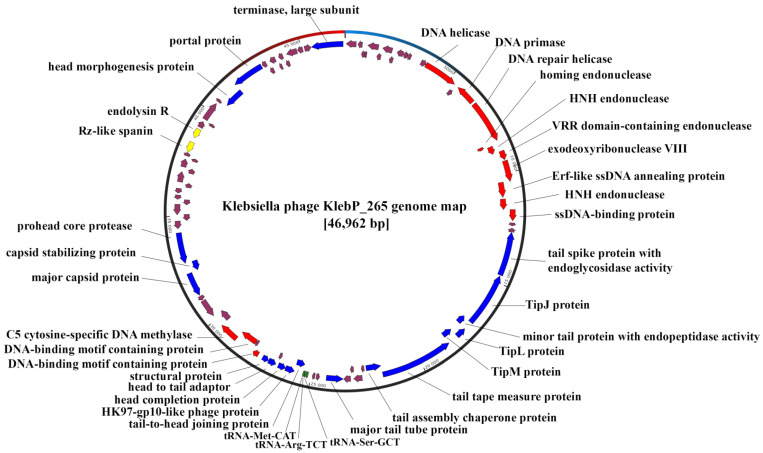
*Klebsiella* phage KlebP_265 genome map. Genes encoding structural proteins and DNA packaging proteins are marked with blue arrows; genes responsible for nucleic acids metabolism are marked with red; genes encoding proteins of lysis cassette are yellow; tRNA genes are green; other genes encoding hypothetical proteins are brown.

**Figure 5 viruses-17-00083-f005:**
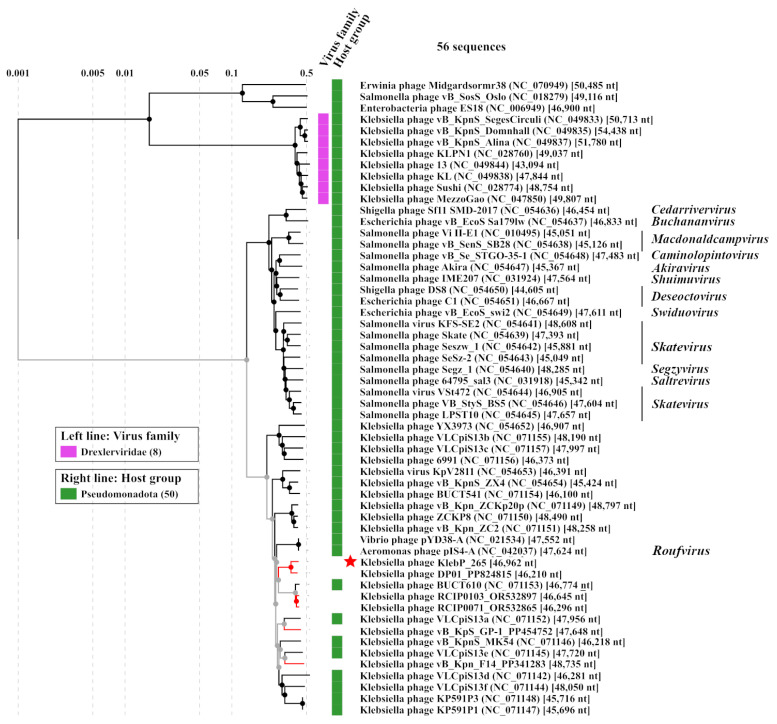
ViPTree analysis for the phage KlebP_265 and related phages. KlebP_265 phage is marked with red asterisk; phage genomes extracted from the NCBI Genbank and added manually into analysis are marked with red lines.

**Figure 6 viruses-17-00083-f006:**
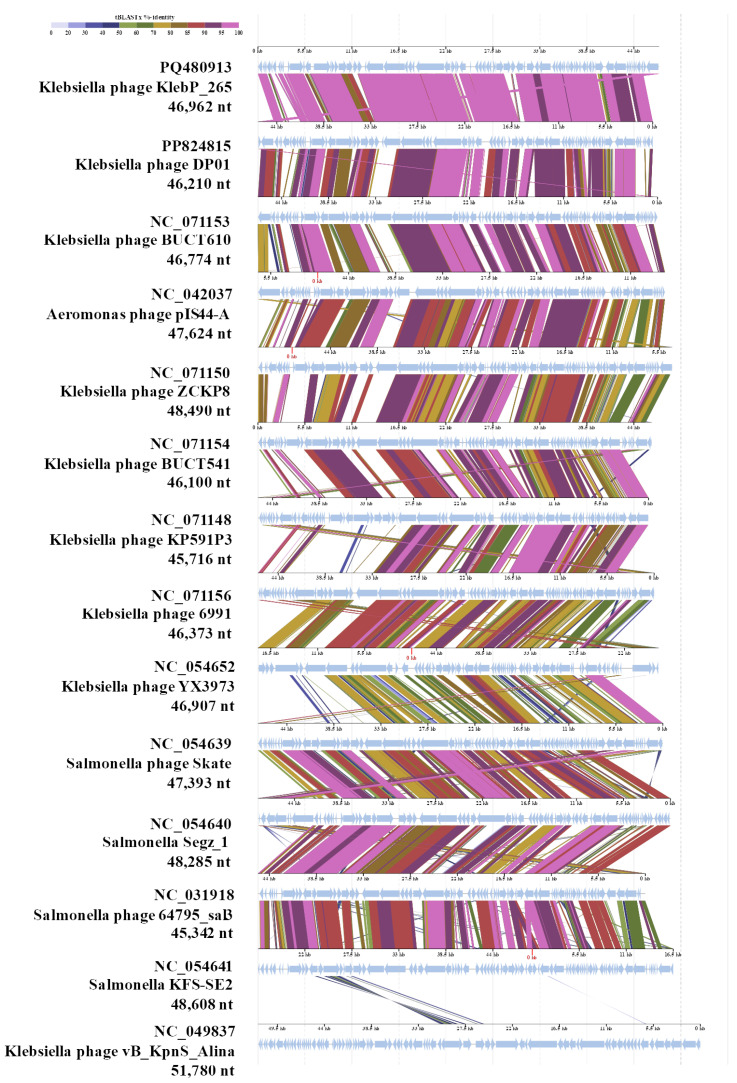
A pairwise comparison between the KlebP_265 and several similar phages performed using the ViPTree tool. *Klebsiella* phage vB_KpnS_Alina (NC_049837) was used as an outgroup.

**Figure 7 viruses-17-00083-f007:**
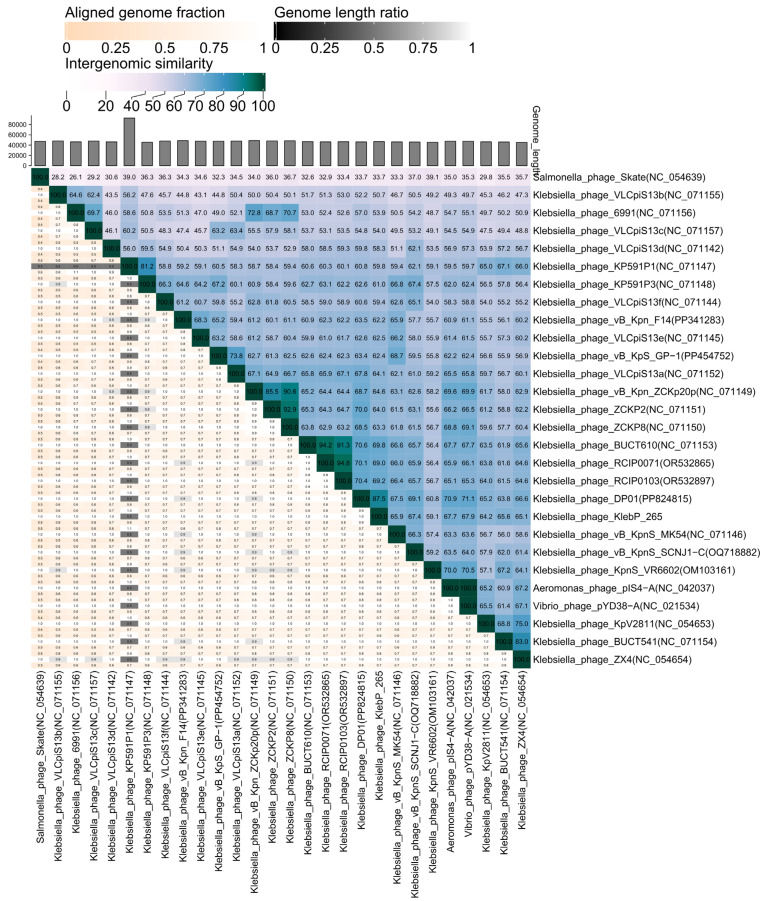
A VIRIDIC heatmap showing the intergenomic similarity within the cluster of KlebP_265 and its most similar phages.

**Figure 8 viruses-17-00083-f008:**
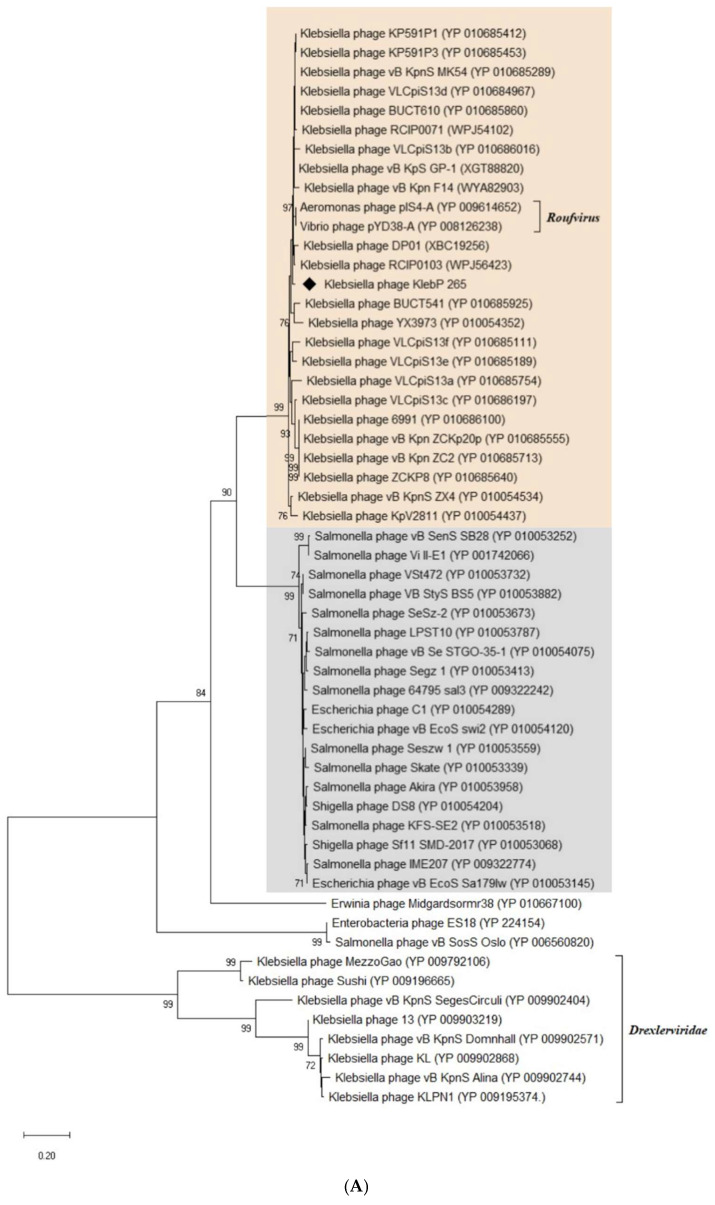
Phylogenetic analysis of the essential proteins of the phage KlebP_265: major tail tubular protein (**A**), large subunit of terminase (**B**). Sequences were aligned using CLUSTALW, phylogenetic tree constructed using MEGA 11.0. Maximum likelihood method with bootstrap 1000 was applied. Protein sequences of phage KlebP_265 are marked with black diamonds.

**Table 1 viruses-17-00083-t001:** Characteristics of *Klebsiella* strains, sensitive to the KlebP_265 phage.

No.	CEMTC Number	Source of Isolation/Date of Isolation	Species, GenBank ID	*wzi* Allele (K-Type), GenBank ID	String Test	Phage Titer, pfu/mL	Relative EOP ^1^	Resistance to Antibiotics ^2^
1	2067	surgical wound/17.02.2015	*K. pneumoniae*, MT439057	2 (K2), MN371471	hpmuc	1 × 10^5^	low	TZP, LEV, CIP, C
2	2291	throat swab, pharyngitis/23.12.2015	*K. pneumoniae*, MT439061	72 (K2), MN371475	hpmuc	1.2 × 10^4^	low	AMC
3	2335	feces, diarrhea/26.01.2016	*K. oxytoca*, MT436848	N.d. ^3^	hpmuc	1.1 × 10^3^	low	S
4	2531	feces, diarrhea/26.04.2016	*K. aerogenes*, MT436839	N.d. ^3^	–	2 × 10^3^	low	AMC
5	2554	feces, diarrhea/05.05.2016	*K. pneumoniae*, MT439068	2 (K2), MN371486	–	9 × 10^3^	low	MDR/AMC, SAM, TZP, CAZ, ATM, CN, LEV, CIP, C
6	2573	feces, diarrhea/24.05.2016	*K. pneumoniae*, MT439069	2 (K2), MN371483	–	1.6 × 10^6^	low	MDR/AMC, SAM, TZP, CAZ, ATM, CN, LEV, CIP, C
7	2945	urine, cystitis/03.05.2017	*K. pneumoniae*, MT439080	2 (K2), MN371498	–	4 × 10^6^	low	MDR/AMC, SAM, CAZ, ATM, CN, LEV, CIP, C
8	3729	oral mucosa/18.07.2019	*K. pneumoniae*, MT489346	2 (K2), MT447747	–	1 × 10^6^	low	S
9	4065	throat swab, pharyngitis/20.11.2020	*K. pneumoniae*, PQ571727	2 (K2), PQ609952	hpmuc	4.5 × 10^11^	high	MDR/AMC, SAM, TIM, FEP, CAZ, FOX, IPM, MEM, ATM, AK, C
10	4087	throat swab, pharyngitis/23.01.2021	*K. pneumoniae*, PQ571728	2 (K2), PQ609953	hpmuc	7 × 10^9^	high	AMC, SAM
11	4090	throat swab, pharyngitis/23.01.2021	*K. pneumoniae*	2 (K2)	hpmuc	3.5 × 10^11^	high	TZP, SAM
12	4117	throat swab, pharyngitis/09.02.2021	*K. pneumoniae*, PQ571729	2 (K2), PQ69954	hpmuc	4 × 10^5^	low	AMC, SAM
13	4123	throat swab, pharyngitis/18.02.2021	*K. pneumoniae*,	2 (K2), PQ679255	–	2.6 × 10^11^	high	CN, LEV
14	4124	feces, diarrhea/18.02.2021	*K. pneumoniae*,	2 (K2), PQ679256	hpmuc	3.4 × 10^11^	high	S
15	4128	throat swab, pharyngitis/02.03.2021	*K. pneumoniae*, PQ571730	2 (K2), PQ609955	–	5.8 × 10^11^	high	S
16	4162	throat swab, pharyngitis/20.05.2021	*K. pneumoniae*, PQ571731	2 (K2), PQ609956	hpmuc	1.1 × 10^8^	medium	S
17	4163	throat swab, pharyngitis/20.05.2021	*K. pneumoniae*, OR544438	2 (K2)	hpmuc	1 × 10^4^	low	CN, AK, CIP
18	4169	throat swab, pharyngitis/27.05.2021	*K. pneumoniae*, PQ671057	72 (K2), PQ609957	–	2 × 10^11^	high	S
**19**	**5232 ^4^**	**sputum sample, pneumoniae/22.04.2022**	***K. pneumoniae*,** **PQ571733**	**2 (K2),** **PQ609958**	**–**	**3.2 × 10^11^**	**1**	**S**
20	5234	sputum sample, pneumoniae/23.04.2022	*K. pneumoniae*	2 (K2)	hpmuc	3.7 × 10^11^	high	S
21	6827	feces, diarrhea/05.10.2022	*K. pneumoniae*, PQ571734	173 (-) ^5^, PQ609959	–	2 × 10^11^	high	MDR/AMC, TIM, TZP, FEP, CAZ, ATM, CN, AK, SXT
22	6846	feces, diarrhea/05.10.2022	*K. pneumoniae*, PQ571735	2 (K2), PQ679259	–	2 × 10^8^	low	MDR/AMC, SAM, TIM, TZP, FEP, CAZ, ATM, IPM, CN, AK, SXT
23	6848	oral mucosa/07.10.2022	*K. pneumoniae*, PQ571736	173 (-) ^5^, PQ609960	–	2 × 10^8^	low	MDR/AMC, SAM, TIM, TZP, FEP, CAZ, ATM, CN, AK, CIP, SXT
24	9609	throat swab, pharyngitis/11.09.2023	*K. pneumoniae*, PQ571737	2 (K2), PQ609961	hpmuc	3 × 10^11^	high	TIM
25	9874	surgical wound/03.10.2023	*K. pneumoniae*, PQ571738	2 (K2), PQ609962	–	4 × 10^4^	low	MDR/AMC, SAM, TZP, FEP, CAZ, CIP, IPM, MEM, DOR, ATM, CN, AK, LEV, CIP, SXT

^1^ The EOP value = phage titer on the tested bacterium/phage titer on host bacterium, EOP values > 0.1 were ranked as ‘high’ efficiency, 0.001–0.1 as ‘medium’ efficiency, below 0.001 as ‘low’ efficiency; ^2^ S—a strain that is sensitive to all tested antibiotics, MDR—strain resistant to three or more classes of antibiotics; ^3^ not determined; ^4^ bacterial host/isolation strain marked with bold; ^5^ K-serotype could not be clearly determined using the *wzi* allele 173 as it corresponds to different K-types (KL102/KL149/KL155).

**Table 2 viruses-17-00083-t002:** Core genes detected using CoreGenes 5.0 and Proteinortho6 in the genomes of the proposed taxonomic units.

No.	Protein (Gene Product ^1^)	Putative *Roufvirinae*		Putative *Skatevirinae*		Putative *Skateviridae*	
		Core Genes 5.0 (24) ^4^	Protein-Otho6 (35) ^5^	Core Genes 5.0 (20) ^4^	Protein-Otho6 (27) ^5^	Core Genes 5.0 (13) ^4^	Protein-Otho6 (20) ^5^
1	HP ^2^ (gp2)		+				
2	HP (gp7)		+				
3	HP (gp9)	+	+				
4	HP (gp10)	+	+				
5	DNA repair helicase (gp16)		+	+	+		+
6	VRR domain-containing protein (gp19)		+		+		+
7	ssDNA-binding protein (gp23)		+				
8	TipJ protein (gp27)	+	+	+	+	+	+
9	Minor tail protein with endopeptidase activity (gp28)		+				
10	TipL protein (gp29)	+	+	+	+	+	+
11	TipM protein (gp30)	+	+	+	+	+	+
12	Tape measure protein (gp31)			+	+		
13	Tail assembly chaperone (gp32)	+	+	+	+	+	+
14	HP (gp33)		+				
15	Major tail tube protein (gp37)	+	+	+	+	+	+
16	Tail-to-head joining protein (gp43)	+	+		+		+
17	HK97-gp10-like protein (gp44)	+	+	+	+	+	+
18	Head completion protein (gp45)	+	+	+	+	+	+
19	Head to tail adaptor (gp47)	+	+	+	+	+	+
20	Structural protein (gp48)		+	+	+		+
21	HP (gp55)		+	+	+		+
22	Major capsid protein (gp56)	+	+	+	+	+	+
23	Capsid stabilizing protein (gp57)			+	+		
24	Prohead core protease (gp58)	+	+	+	+	+	+
25	HP (gp59)	+	+				
26	HP (gp60)						
27	HP (gp61)	+	+				
28	Lar-like restriction alleviation protein (gp62)				+		
29	HP (gp63)				+		
30	HP (gp64)	+	+				
31	HP (gp65)	+	+		+		+
32	DUF6378 domain-containing protein (gp66)	+	+				
33	Endolysin R (gp72)	+	+		+		+
34	HP (gp73)	+	+				
35	HP (gp74)		+				
36	Head morphogenesis protein (gp77)	+	+	+	+	+	+
37	Portal protein (gp78)	+	+	+	+	+	+
38	InsA C-terminal domain protein (gp86)	+	+	+	+	+	+
39	Phosphatase (gp87)	+	+				
40	Lipoprotein (YP_010053329.1) ^3^				+		
41	Holin (YP_010053300.1) ^3^			+	+		
42	Endolysin (YP_010053299.1) ^3^			+	+		

^1^ Numeration of gene products is according to the position of the corresponding genes in the genome of the KlebP_265 phage; ^2^ HP—hypothetical protein; ^3^ protein IDs according to the GenBank annotation of the phage Skate (NC_054639); ^4^ the number of conservative genes identified using Core Genes 5.0 is given in parentheses; ^5^ the number of conservative genes identified using Proteinortho6 is given in parentheses. Yellow boxes represent core genes in the *Roufvirinae* subfamily; brown boxes represent core genes in the *Skatevirinae*; red boxes represent core genes in the *Skateviridae* family.

## Data Availability

The KlebP_265 genome sequence was deposited in GenBank under the accession number PQ480913.

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
