# Peer review of "Phage vB_KlebPS_265 Active Against Resistant/MDR and Hypermucoid K2 Strains of Klebsiella pneumoniae"

_viruses, 2025, doi:10.3390/v17010083_

Round 1

Reviewer 1 Report

Comments and Suggestions for Authors

The presented original article by Yakubovskij et al. „Phage vB_KlebPS_265 active against Resistant/MDR and  Hypermucoid K2 Strains of Klebsiella pneumoniae“ is a very well-conducted and written research of detection of a new bacteriophage with the lytic activity to K2 types of Klebsiella pneumoniae. Congratulations to the authors on well design of the study.

A few remarks or suggestions for the authors.

In the abstract K.pneumoniae infection and AMR

1. In the Introduction after the sentence regarding hvKp (line 61) please add the comment about molecular detection of hypervirulent K.pneumoniae and the scoring system for determination of these strains. (https://www.nature.com/articles/s41467-021-24448-3#Tab1 ; https://github.com/klebgenomics/Kleborate )

2. - 2.1. Please add information regarding the origin of the sample, what is the diagnosis of the patient/type of K.pneumoniae infection (known only from the abstract). It might be important as both the strain and the phage were isolated from the same sample from the same patient.

Regarding the characteristics of the origin K.pneumoniae strain please add a line in results on its type and resistance in the context of its relation with the phage activity and relations with other strains on which the detected phage gave the activity.

As You mentioned antibiotic susceptibility testing of K.pneumoniae in the Methods, please add a table in relation of antibiotic resistance of tested trains and the activity of detected phage (on % of carbapenem, quinolone, etc resistant strains). It would provide an additional result worth a line or two in the discussion.

Minor remarks:

Line 45 – Introduced an abbreviation of cKP, but from line 57 it is cKp. Also, hvKp is introduced, but in line 546/7 there is hmKp, likewise hmK2 (line 542,543,..)or K2hm (line 590). Please unify throughout of text.

Line 529 – Add the reference to the comment on K2 being common in the world.

Author Response

Answer to Reviewer 1

The presented original article by Yakubovskij et al. „Phage vB_KlebPS_265 active against Resistant/MDR and  Hypermucoid K2 Strains of Klebsiella pneumoniae“ is a very well-conducted and written research of detection of a new bacteriophage with the lytic activity to K2 types of Klebsiella pneumoniae. Congratulations to the authors on well design of the study.

We are grateful to the reviewer for the positive assessment of our work and for the valuable comments that we think will help us improve our manuscript.

A few remarks or suggestions for the authors.

  1. In the Introduction after the sentence regarding hvKp (line 61) please add the comment about molecular detection of hypervirulent K.pneumoniaeand the scoring system for determination of these strains. (https://www.nature.com/articles/s41467-021-24448-3#Tab1 ; https://github.com/klebgenomics/Kleborate )

The suggested information was added to the Introduction. Lines 59-61.

  1. - 2.1. Please add information regarding the origin of the sample, what is the diagnosis of the patient/type of K.pneumoniaeinfection (known only from the abstract). It might be important as both the strain and the phage were isolated from the same sample from the same patient.

Data was added into the Table 1.

Regarding the characteristics of the origin K.pneumoniae strain please add a line in results on its type and resistance in the context of its relation with the phage activity and relations with other strains on which the detected phage gave the activity.

The information was added. Lines 350-353.

As You mentioned antibiotic susceptibility testing of K.pneumoniae in the Methods, please add a table in relation of antibiotic resistance of tested trains and the activity of detected phage (on % of carbapenem, quinolone, etc resistant strains). It would provide an additional result worth a line or two in the discussion.

The data was added to the Table 1

Minor remarks:

Line 45 – Introduced an abbreviation of cKP, but from line 57 it is cKp. Also, hvKp is introduced, but in line 546/7 there is hmKp, likewise hmK2 (line 542,543,..)or K2hm (line 590). Please unify throughout of text.

cKP was corrected to cKp. We agree with the reviewer that “hypermucous” is not equal to “hypervirulent”. Therefore, the abbreviation hm (Hypermucous) phenotype was introduced in methods Line 172. K2hm was corected to hmK2. Line 604.

Line 529 – Add the reference to the comment on K2 being common in the world.

Phrase was corrected and references added. Lines 538-539

Reviewer 2 Report

Comments and Suggestions for Authors

The work of Yakubosvskij et al., focuses on the characterization of a lytic phage vB_KlebPS_265 isolated from a clinical sputum sample. The authors perform both phenotypic (adsorption curve, one-step curve, multi-killing curve, pH, and temperature stability test) and genomic characterization (phage genome sequencing, phage genome annotation, and comparative genomic study together with taxonomic characterization). In general, the research is well structured and developed, however, I would like to propose some comments in order to improve the work. 

Abstract 

1.     The abstract does not reflect the structure of introduction, material and methods, results, and conclusion. It is an exposition of the results obtained. Perhaps it should be rewritten to comply with the journal format.

2.     Line 14 replaces «  K pneumoniae »  for « K. pneumoniae »

Introduction

3.     Line 32-33: Rephrase this sentence, one option could be: “The genus Klebsiella is included in the gram-negative bacilli”.

4.     Line 75-78: The characteristics of the three types of phages belonging to the Caudoviral phages should be explained here.

Material and methods

5.     I would recommend making a table with the primers used throughout the manuscript and removing them from the text to make it easier to read.

6.     Line 118: In this sentence when referring to “excised from agar”. How exactly did you do it, i.e. did you pick the phage plates or cut the Agar and leave the phage in the phage buffer? I think it would be interesting to clarify it.

7.     Line 141: I think you should give a little more information about the strain collection used.

8.     Line 141: Replace « Spot-assay » with « Spot test assay »

9.     Line 170: Which antimicrobial has been tested should be indicated in this section as they only appear in supplementary material.

10.  Line 173: Section Biological properties of Phage KlebP_265. I think you should divide this section into several subsections (Adsorption curve, one-step growth curve, multi-killing curve, and lysogenic study).

11.  Line 175: Remove the sentences « Three technical repeats each »

12.  Line 180: To determine the latency time and burst size, the adsorption curve is not performed but the one-step growth curve. Indicate in the text.

13.  Line 188: Change “to calculate” to “to determine”

14.  Line 188: Remove « In the life cycle of the »

15.  Line 196: For how long the co-culture assay was performed to see if the phages were present in the remaining bacteria in a lysogenic manner. Please specified

16.  Line 195-205: I do not quite understand the experiment. You are testing those bacteria that are resistant to the phage, but the resistance may be due to multiple factors of defense of the bacteria against the phage, not only due to the integration of the phage itself in the genome. Furthermore, if the bacteria are resistant, the phage, even if it has lysogenic capacity, cannot integrate. Should not you infect the strain in liquid medium at an early logarithmic phase, and collect the culture after a few hours of infection (2-4h), centrifuge it to keep the bacteria and seed them, and then perform PCR?

17.  Line 225: Add a sentence introducing DNA extraction. On the other hand, DNA extraction by the phenol:chloroform:isoamylalchol method is a standardized protocol that should be cited as such.

18.  Line 247: Has the HHmer program been used to annotate the phages?

Results:

19.  Line 281: It should indicate the characteristics of phages with Siphovirus morphology.

20.  Figure 1 (A): You should put a scale on the image

21.  Line 289-290: The sentence beginning with “This suggest...” should be deleted as it should belong in the discussion section. As a result, you only have to expose the results obtained from your work.

22.  Line 290: Replace « One-step growth » for « One-step growth curve »

23.  Line 291: Replace « minutes » for « min »

24.  Figure 2: (A) The scale of the curve must be wrong. Should not it be percentages of free phages? Starting with 100% at the top on the Y-axis?

(B) I would put the Y axis in logarithmic scale. In addition, I think you should indicate in this figure with arrows the latency period as well as the burst size.

(C) In figure C, I would put the MOI as 0.01, 0.1, and 1. Also, it seems that the scale is wrongly set to PFU/mL since otherwise, the control value would have to be 0 PFU/mL as there are no phage infections.

25.  Line 307-308: « Including …. » This sentence is not well understood. How many strains of Klebsiella were used? How many strains of K. pneumoniae were infected?

26.  Line 313: « which indicates … » This sentence has to be removed from the results and added to the discussion.

27.  Line 316-317: Rephrase this sentence, it is not well understood.

28.  Line 318-319; Line 331-332: « The K2 capsule … » and « It appears …» These sentences have to be removed from the results and added to the discussion.

29.  Line 335: Same comment as in material and methods, which antibiotics were tested?

30.  Line 355: I think I should add a graph to show the pH values.

31.  In general, the curves that include different conditions (Multi-kiling curve, Stability), being in black and white, make it a little difficult to locate the different conditions on the graph.

32.  Line 386-389; 396-398; 401-402; 453-455: These sentences have to be removed from the results and added to the discussion.

33.  Line 459: There should be no citations as only your results should be exposed .

34.  Figure 6: This genomic comparison only shows the similarity in a paired form, i.e. of KlebP_265 with DP01, DP01 with BUCT610, etc... Therefore I think that only the comparison with DP01 should be kept.

Discussion

35.  Line 521: Add a brief introduction to the discussion

36.  Line 543-547: Are there any papers that support these two hypotheses?

37.  Line 550: Explain the relevance of this gene.

38.  Line 578, line 584, line 586: Add a reference that confirm these statements

Author Response

Answer to Reviewer 2

The work of Yakubosvskij et al., focuses on the characterization of a lytic phage vB_KlebPS_265 isolated from a clinical sputum sample. The authors perform both phenotypic (adsorption curve, one-step curve, multi-killing curve, pH, and temperature stability test) and genomic characterization (phage genome sequencing, phage genome annotation, and comparative genomic study together with taxonomic characterization). In general, the research is well structured and developed, however, I would like to propose some comments in order to improve the work. 

We thank the reviewer 2 for the positive response and tried to make corrections, which we believe will improve our manuscript. 

Abstract 

  1. The abstract does not reflect the structure of introduction, material and methods, results, and conclusion. It is an exposition of the results obtained. Perhaps it should be rewritten to comply with the journal format.

We are grateful to the reviewer for this comment, but the abstract contains not only results, but also brief introduction and conclusion. At the same time, methods used to isolate and characterize the phage were routine and standard, so we consider it unreasonable to add them to the abstract.

  1. Line 14 replaces « K pneumoniae »  for « K. pneumoniae »

Was corrected.

Introduction

  1. Line 32-33: Rephrase this sentence, one option could be: “The genus Klebsiella is included in the gram-negative bacilli”.

Was rephrased.

  1. Line 75-78: The characteristics of the three types of phages belonging to the Caudoviral phages should be explained here.

 Was added. Lines 76-78

Material and methods

  1. I would recommend making a table with the primers used throughout the manuscript and removing them from the text to make it easier to read.

We thank the reviewer for this comment, but we prefer to give the sequences of primers in the text. The study used only three pairs of primers, which is not that many.

  1. Line 118: In this sentence when referring to “excised from agar”. How exactly did you do it, i.e. did you pick the phage plates or cut the Agar and leave the phage in the phage buffer? I think it would be interesting to clarify it.

Was clarified. Line 123

  1. Line 141: I think you should give a little more information about the strain collection used.

Information was added into Table 1.

  1. Line 141: Replace « Spot-assay » with « Spot test assay »

Was corrected. Line 146

  1. Line 170: Which antimicrobial has been tested should be indicated in this section as they only appear in supplementary material.

The data was added to the Table 1, and in Methods, section 2.6

  1. Line 173: Section Biological properties of Phage KlebP_265.I think you should divide this section into several subsections (Adsorption curve, one-step growth curve, multi-killing curve, and lysogenic study).

We have divided the section into paragraphs, according to experiments described.

  1. Line 175: Remove the sentences « Three technical repeats each »

Was removed

  1. Line 180: To determine the latency time and burst size, the adsorption curve is not performed but the one-step growth curve. Indicate in the text.

Was done. Lines 192-193

  1. Line 188: Change “to calculate” to “to determine”

Was changed. Line 201

  1. Line 188: Remove « In the life cycle of the »

Was removed. Line 201

  1. Line 196: For how long the co-culture assay was performed to see if the phages were present in the remaining bacteria in a lysogenic manner. Please specified

Co-culturing was performed for 19-20 hours. Then any colonies of bacteria, that grew inside the clear, large plaques formed with KlebP_265 were transferred onto NA and tested for their sensitivity to the KlebP_265 phage. To purify BIMs from residual phage particles, they were passaged three times on the plates containing NA, and then used to detect KlebP_265 DNA. Lines 207-217

  1. Line 195-205: I do not quite understand the experiment. You are testing those bacteria that are resistant to the phage, but the resistance may be due to multiple factors of defense of the bacteria against the phage, not only due to the integration of the phage itself in the genome. Furthermore, if the bacteria are resistant, the phage, even if it has lysogenic capacity, cannot integrate. Should not you infect the strain in liquid medium at an early logarithmic phase, and collect the culture after a few hours of infection (2-4h), centrifuge it to keep the bacteria and seed them, and then perform PCR?

We agree with the reviewer that there are various mechanisms of resistance to phages, and not just the lysogenic one. However, our experiments have shown that there is no phage DNA present in the surviving bacteria, and none of the isolates were lysogenic. We thank the reviewer for suggesting the scheme of the experiment using liquid culture, and we will try to implement it.

  1. Line 225: Add a sentence introducing DNA extraction. On the other hand, DNA extraction by the phenol:chloroform:isoamylalchol method is a standardized protocol that should be cited as such.

Introduction to the method with reference was added. Line 237

  1. Line 247: Has the HHmer program been used to annotate the phages?

Thanks, will use it next time. Here the HHPred algorithm was used. 

Results:

  1. Line 281: It should indicate the characteristics of phages with Siphovirus morphology.

The characteristics of siphovirus morphotype were added into introduction. Lines 76-78

  1. Figure 1 (A): You should put a scale on the image

Was added

  1. Line 289-290: The sentence beginning with “This suggest...” should be deleted as it should belong in the discussion section. As a result, you only have to expose the results obtained from your work.

Was removed

  1. Line 290: Replace « One-step growth » for « One-step growth curve »

Was corrected. Line 304

  1. Line 291: Replace « minutes » for « min »

Was corrected. Line 305

  1. Figure 2: (A) The scale of the curve must be wrong. Should not it be percentages of free phages? Starting with 100% at the top on the Y-axis?

Thanks the reviewer for comment. Not, the scale reflect the number of free phages, which form plaques. It is one of the possible ways to descibe the data.

(B) I would put the Y axis in logarithmic scale. In addition, I think you should indicate in this figure with arrows the latency period as well as the burst size.

Latent period and burst size were added to Figure 2B

(C) In figure C, I would put the MOI as 0.01, 0.1, and 1. Also, it seems that the scale is wrongly set to PFU/mL since otherwise, the control value would have to be 0 PFU/mL as there are no phage infections.

We a grateful to the reviewer for this comment. It is a mistake, it should be CFU, as the bacterial cells are counted. Was corrected.

  1. Line 307-308: « Including …. » This sentence is not well understood. How many strains of Klebsiellawere used? How many strains of K. pneumoniae were infected?

The sentences have been rephrased in order to make them more readable. Lines 320-323

  1. Line 313: « which indicates … » This sentence has to be removed from the results and added to the discussion.

Was removed

  1. Line 316-317: Rephrase this sentence, it is not well understood.

Was re-phrased. Lines 330-332

  1. Line 318-319; Line 331-332: « The K2 capsule … » and « It appears …» These sentences have to be removed from the results and added to the discussion.

Was removed from results

  1. Line 335: Same comment as in material and methods, which antibiotics were tested?

Data was added into Table 1 and to the methods, section 2.6

  1. Line 355: I think I should add a graph to show the pH values.

We thank the reviewer for this comment, but prefer to put this data into the text.

  1. In general, the curves that include different conditions (Multi-kiling curve, Stability), being in black and white, make it a little difficult to locate the different conditions on the graph.

We tried to make them colored, but this did not significantly affect the situation, because in Figure 3b the stability curves are too close to each other. For a better understanding, these data are described in the text.

  1. Line 386-389; 396-398; 401-402; 453-455: These sentences have to be removed from the results and added to the discussion.

Was removed from the text.

  1. Line 459: There should be no citations as only your results should be exposed .

We appreciate the reviewer's comment, but this section contains comparative genomic analysis. Therefore, citations are valuable to the readers to understand the data.

  1. Figure 6: This genomic comparison only shows the similarity in a paired form, i.e. of KlebP_265 with DP01, DP01 with BUCT610, etc... Therefore I think that only the comparison with DP01 should be kept.

We are grateful for the reviewer's comment, but prefer our variant of the Figure 6. It is important to demonstrate the similar genome organization for the proposed subfamily.

Discussion

  1. Line 521: Add a brief introduction to the discussion

Was Added. Lines 528-529

  1. Line 543-547: Are there any papers that support these two hypotheses?

Thanks for the reviewer for this comment. We tried to explain our results in discussinon, we didn’t find the publications to support this hypothesis. It needs further investigation.

  1. Line 550: Explain the relevance of this gene.

Was added. Lines 560-563

  1. Line 578, line 584, line 586: Add a reference that confirm these statements

Were added. Lines 590, 596, 598